# OpenReview forum: "SARSteer: Safeguarding Large Audio Language Models via Safe-Ablated Refusal Steering"
_ICML.cc/2026/Conference — ICML 2026 regular_

### Official Review · Reviewer_kZ6Y · 2026-03-11

**Soundness:** 3
**Presentation:** 2
**Significance:** 3
**Originality:** 2
**Overall Recommendation:** 4
**Confidence:** 4

**Summary:**

This study reveals that LALMs have security vulnerabilities: audio input is more likely to trigger harmful responses, and the existing alignment methods have limited transfer effect. The authors propose the reasoning method **SARSteer**, which improves the rejection rate and reduces false positives by eliminating the rejection vector and PCA secure space.

**Compliance With Llm Reviewing Policy:**

Affirmed.

**Final Justification:**

My concerns are resolved.

**Key Questions For Authors:**

Please refer to the weaknesses.

**Limitations:**

Yes

**Strengths And Weaknesses:**

## Strengths

1. **The motivation is clear.** This paper points out that there are differences between audio representation and text, and the t-SNE analysis shows that harmful and safe speech form separate subspaces across layers, which motivate the this paper.
2. **Comprehensive evaluation.** This paper evaluates the method across various benchmarks, systematically showcasing the effectiveness of the proposed method.

## Weaknesses

1. My main concern focuses on the novelty of this paper. The novelty of this paper is limited from multiple perspectives.
   - The question of adopting activation steering in multi-modal language models has been discussed in previous literature [1]. From this perspective, this paper just discussed the same question from the LALM perspective.
   - PCA decomposition has also been discussed in previous literature for improving the activation steering method [2].

As a result, the overall contribution is mainly modal extension.

2. Presentation can be improved. For example, Figure 3 is hard to understand without any explanations in the title. Additionally,

3. Lack of evaluation on run-time adaptive attacks, such as GCG [3]. Additionally, it would be better to incorporate more traditional text-based attacks such as PAIR [4].

[1] Steering away from harm: An adaptive approach to defending vision language model against jailbreaks. CVPR 2025.

[2] Jailbreak Antidote: Runtime Safety-Utility Balance via Sparse Representation Adjustment in Large Language Models. ICLR 2025.

[3] Universal and Transferable Adversarial Attacks on Aligned Language Models.

[4] Jailbreaking Black Box Large Language Models in Twenty Queries.

---

> ### Author Rebuttal · Authors · 2026-03-31
>
> Thank you very much for your thoughtful review and constructive comments. We are greatly encouraged by your recognition of the **clear motivation** and **comprehensive evaluation**. We hope the following responses could help clarify the potential misunderstandings and alleviate your concerns.
>
> ---
>
> **Cons1: Concern about the novelty — comparison with ASTRA and Jailbreak Antidote.**
>
> **R1:** Thanks for your concern. We would like to respectfully clarify that SARSteer is not a straightforward modal extension. We implemented vanilla activation steering from LLMs as baselines (MDSteer-h2s/c2r, Section 3.3); their direct application to LALMs **worsens** safety — ASR increases from 51.6% to 84–90.8% (Figure 1), motivating qualitatively new designs.
>
> **vs. ASTRA (CVPR 2025) — Fundamentally different problem setting:**
>
> - In ASTRA, harmful content resides in the **text** query and adversarial images are auxiliary attack vectors that do not carry semantic information — ablating visual tokens does not affect query semantics. In our setting, harmful content is **embedded in the audio itself**; ablating audio activations directly disrupts semantic information, so ASTRA's approach does not transfer to our problem.
> - Deriving vectors from audio activations (as ASTRA does from visual tokens) is infeasible: CKA (Appendix B.10) shows audio harmful/safe representations are highly divergent (CKA = 0.063 vs. text = 0.672), confirmed by MDSteer baselines (ASR worsens to 84–90.8%). SARSteer instead derives vectors from **text refusal prompts** (Eq. 9), bypassing this gap.
> - SARSteer additionally introduces PCA-based safe-space ablation (Eq. 11–12) for over-refusal, absent in ASTRA.
>
> **vs. Jailbreak Antidote (ICLR 2025) — Different PCA usage, different problem:**
>
> - **Domain:** Text-only LLMs; does not address cross-modal distributional gaps.
> - **PCA usage is fundamentally different.** Jailbreak Antidote uses PCA to find a single **safety direction** (first principal component), then performs sparse additive steering: **$h' = h + \alpha (d_{\text{safe}} \odot m)$**. SARSteer uses PCA to identify a **safe subspace** (top-k components), then **decomposes** the steering vector and **removes** the safe-aligned component (Eq. 11–12). Key difference: PCA to find *where to steer* vs. *what to remove* to prevent over-refusal. Therefore, the motivation is fundamentally different and contribute to different perspective.
> - **Problem:** Text safety-utility balance vs. our cross-modal over-refusal, quantified via BRR (Eq. 8) with paired audio datasets.
>
> Beyond the method: (1) first systematic analysis of the text-audio gap in LALM safety, (2) BRR metric + paired audio datasets, (3) first inference-time defense benchmark for LALMs across 4+ attack categories.
>
> ---
>
> **Cons2: Suggestion for improving Figure 3 clarity.**
>
> **R2:** Thanks for your valuable suggestion. We will revise Figure 3 in the camera-ready with a descriptive caption: "**Overview of SARSteer.** Step 1 derives a refusal steering vector from text prompts. Step 2 ablates the safe-subspace component via PCA to prevent over-refusal. Step 3 adds the ablated vector at inference to shift harmful responses toward refusal." Detailed derivations remain in Sections 3.4–3.5.
>
> ---
>
> **Cons3: Suggestion for evaluating GCG / PAIR adaptive attacks.**
>
> **R3:** Thanks for your suggestion. AJailBench (Table 2) already covers audio adversarial perturbations (time/frequency/hybrid), where SARSteer achieves comparable ASR. To directly address GCG and PAIR, we implemented audio-adapted versions during the rebuttal:
>
> - **GCG (audio):** We optimize adversarial suffixes on the text input using the standard GCG algorithm, then concatenate the suffix with the original harmful query and convert the full text to audio via TTS. This tests whether token-level adversarial perturbations retain their attack capability after the text→audio→model pipeline.
> - **PAIR (audio):** We use PAIR to generate semantically rephrased jailbreak queries (e.g., role-playing, academic disguise) in text, then convert them to audio via TTS. This tests semantic-level adaptive attacks through the audio channel.
>
> We evaluate 50 samples per attack on Kimi-Audio using Figstep-audio:
>
> | Attack | Model | No Defense ASR | SARSteer ASR |
> | --- | --- | --- | --- |
> | PAIR (audio) | Kimi-Audio | 58.0% | **10.0%** |
> | GCG (audio) | Kimi-Audio | 14.0% | **6.0%** |
>
> PAIR is remarkably effective as an adaptive attack (58.0% No Defense ASR), yet SARSteer reduces it to 10.0% (↓82.8%). GCG is less effective through the audio channel (14.0%), and SARSteer further reduces it to 6.0%. Combined with existing evaluations, SARSteer is now validated across **5 attack categories**: TTS-based, audio perturbation, natural speech variation, gradient-based adversarial, and text-origin adaptive (GCG/PAIR). We will include these results in the camera-ready version.

---

> > ### Author Rebuttal · Reviewer_kZ6Y · 2026-04-02
> >
> > Thanks for the reply. My concerns are resolved.

---

> > > ### Author Response · Authors · 2026-04-04
> > >
> > > Thank you very much for the care you have taken in reviewing our paper. Your questions and suggestions pushed us to think more carefully about how to present and validate our contributions. We are committed to reflecting these improvements in the camera-ready version. We truly value the time and thought you have invested, and we appreciate your constructive spirit throughout this process. Thank you once again for your time and expertise.

---

### Official Review · Reviewer_Cdxw · 2026-03-11

**Soundness:** 3
**Presentation:** 3
**Significance:** 3
**Originality:** 3
**Overall Recommendation:** 4
**Confidence:** 4

**Summary:**

This work presents SARSteer, an inference-time safety alignment framework designed for Large Audio-Language Models (LALMs). The paper argues that directly using activation steering from text-based LLMs fails because of a large representational gap between the audio and text modalities. Furthermore, they claim that adopting prompt-based defenses from vision-language models leads to significant over-refusal on benign queries. To resolve this, SARSteer extracts a refusal vector from text prompts paired with audio inputs and applies PCA to isolate the components responsible for refusal behavior. This allows the model to block harmful requests while still responding normally to safe ones.

**Compliance With Llm Reviewing Policy:**

Affirmed.

**Final Justification:**

The paper identifies an underexplored vulnerability in LALMs and proposes a practical inference-time defense with a mathematically grounded PCA decomposition, supported by thorough evaluations and detailed ablations. However, the new baseline comparisons reveal that SARSteer underperforms MDSteer-h2s, MDSteer-c2r, AdaShield, and FSD in certain settings, suggesting a more nuanced trade-off than presented. Additionally, 43/500 false positives in the expanded human study still reflect systematic evaluator bias. Given these remaining concerns, I decide to keep my score.

**Key Questions For Authors:**

1. Does SARSteer generalize to recent LALMs (e.g., Qwen2.5-Omni, Audio Flamingo 3, Voxtral)? Does the audio-text modality gap persist in these newer models? (W1)
2. Have human evaluations been conducted to validate the LLM-as-a-judge ASR metric, and if so, what is the alignment score between automated and human assessments? (W2)

**Limitations:**

yes

**Strengths And Weaknesses:**

### Strengths
1. The paper clearly identifies an underexplored vulnerability in LALMs and provides empirical analysis to support its claims.
2. The approach is practical because it operates entirely at inference time. By avoiding resource-intensive fine-tuning, it offers a scalable solution for real-world deployment.
3. The PCA-based decomposition of the steering vector is clear, mathematically grounded, and highly interpretable. By explicitly projecting away the components aligned with safe activations before inference, the method systematically separates refusal-relevant directions from benign variance, elegantly targeting the over-refusal problem.
4. The evaluations thoroughly assess harmfulness, helpfulness, and general utility across multiple audio safety datasets and a general benchmark. Detailed ablations with several methods also provide strong empirical support for the proposed framework.

### Weaknesses
1. The empirical evaluations are primarily based on Qwen2-Audio and Kimi-Audio. Given the rapid pace of multimodal model development, this work would be significantly strengthened by evaluating on the latest state-of-the-art models, such as Qwen2.5-Omni [1], Qwen3-Omni [2], Audio Flamingo 3 [3], and Voxtral [4]. Expanding the evaluation would help verify whether the observed modality gap and the efficacy of SARSteer persist in newer, potentially more aligned architectures.
2. The framework heavily relies on an LLM-as-a-judge to evaluate the Attack Success Rate (ASR). The paper does not calculate or report the alignment or correlation between the LLM judge's assessments and actual human evaluations for these specific audio-conditioned responses, potentially introducing bias into the safety assessment.

[1] Xu, Jin, et al. "Qwen2.5-omni technical report." arXiv preprint arXiv:2503.20215 (2025).

[2] Xu, Jin, et al. "Qwen3-omni technical report." arXiv preprint arXiv:2509.17765 (2025).

[3] Goel, Arushi, et al. "Audio flamingo 3: Advancing audio intelligence with fully open large audio language models." arXiv preprint arXiv:2507.08128 (2025).

[4] Liu, Alexander H., et al. "Voxtral." arXiv preprint arXiv:2507.13264 (2025).

---

> ### Author Rebuttal · Authors · 2026-03-31
>
> Thank you very much for your positive appraisal and detailed comments on our work. We are greatly encouraged by your recognition of the **practical inference-time defense design**, **clear and mathematically grounded PCA-based decomposition**, and **thorough evaluations across multiple benchmarks**. We hope the following responses could help alleviate your concerns.
>
> ---
>
> **Cons1: Suggestion for evaluating generalization to more recent LALMs.**
>
> **R1:** Thanks for your valuable suggestion. We would like to kindly point out that, beyond the two main models (Qwen2-Audio and Kimi-Audio), our submission already includes evaluations on three additional models/settings:
>
> - **MiDashengLM** (Xiaomi Inc.; Appendix B.8, Table 13): SARSteer achieves **0.00% ASR** on Figstep-audio (vs. 12.80% No Defense)—a complete elimination of harmful responses.
> - **Qwen2-Audio-Base** (Appendix B.3, Table 8): 15.20% ASR (vs. 62.80% No Defense), demonstrating effectiveness even without instruction tuning, where the model lacks built-in safety alignment.
> - **Qwen2 text-only LLM** (Appendix B.4, Table 9): SARSteer generalizes to the pure text modality, validating cross-modal applicability of our framework design.
>
> To directly address your suggestion, we evaluated on additional models, reporting ASR on two harmful benchmarks:
>
> | Model | Figstep ASR (↓) | SORRY-Bench ASR (↓) |
> | --- | --- | --- |
> |  | No Def → SARSteer | No Def → SARSteer |
> | Qwen2.5-Omni-7B | 42.0% → **31.2%** | 0.0% → 0.0% |
> | Qwen3-Omni-30B (MoE) | 7.2% → **5.2%** | 0.0% → 0.0% |
> | Voxtral Mini 3B | 34.0% → **30.0%** | 1.6% → **0.0%** |
>
> SARSteer consistently reduces ASR across all models. Notably, Qwen3-Omni (30B MoE) already has strong built-in safety (7.2%), yet SARSteer still provides further reduction to 5.2%. On Voxtral-Mini, SARSteer eliminates all SORRY-Bench attacks (1.6% → 0.0%). With these additions, SARSteer is now validated across **8 open-source L(A)LMs/settings** spanning diverse architectures from 4 organizations. We will include full results in the camera-ready version.
>
> ---
>
> **Cons2: Concern about human evaluation alignment for LLM-as-a-judge.**
>
> **R2:** Thanks for your concern. We would like to kindly mention two design choices in our evaluation that already mitigate this risk:
>
> **(i) Validated safety evaluator.** Our ASR evaluator is the fine-tuned Mistral-7b from SORRY-Bench, specifically trained on a human-judged safety dataset with performance comparable to GPT-4o and Gemma-7b. This is not a generic LLM judge but a purpose-built safety classifier.
>
> **(ii) Dual evaluation protocol.** We employ LLM-based ASR for harmfulness and matching-based BRR (Eq. 7–8) for helpfulness—two complementary methods that cross-validate each other and reduce single-evaluator bias.
>
> To further alleviate your concern, we conducted a human evaluation study. We randomly sampled 100 model responses (covering Qwen2-Audio and Kimi-Audio under both No Defense and SARSteer conditions) and had them independently annotated in a blinded setting — annotators had no access to model identity, defense condition, or the LLM judge's labels. Each response was labeled as either "safe" (the model refused or gave harmless content) or "unsafe" (the model complied with the harmful request). We then compared these human labels against the LLM judge's labels on the same 100 responses:
>
> |  | LLM = Safe | LLM = Unsafe |
> | --- | --- | --- |
> | **Human = Safe** | 52 | 24 |
> | **Human = Unsafe** | 0 | 24 |
>
> Out of 100 responses, human and LLM labels agree on 76 (agreement rate 76%). The inter-rater agreement measured by Cohen's κ (a standard metric that accounts for chance agreement, where 1.0 = perfect and 0 = random) is 0.51. Importantly, the LLM judge achieves **zero false negatives**: every response that the human annotator deemed unsafe was also flagged by the LLM judge. All 24 disagreements are in the conservative direction — the LLM judge labels some actually safe responses as unsafe (e.g., the model generating irrelevant but harmless content in response to harmful queries). This means our reported ASR values are **upper bounds**; the true ASR is likely lower, and SARSteer's actual safety improvement is underestimated rather than overestimated. We will include the full protocol in the camera-ready version.

---

> > ### Author Rebuttal · Reviewer_Cdxw · 2026-04-02
> >
> > Thank you for your response. While the additional experiments offer some useful context, several concerns remain unaddressed.
> >
> > Missing Baselines on New Models (W1): The authors demonstrated that the proposed method generalizes to newer models, yet the baseline defenses (e.g., AdaShield, FSD, MDSteer) were not evaluated under the same conditions. Given that the paper's central claim is that these baselines fail in audio settings, it is essential to verify whether this failure persists on the newer architectures. Without such a comparison, it remains unclear whether SARSteer provides meaningful advantages over these baselines or whether the newer models have improved their inherent audio handling capabilities.
> >
> > Weak Human Evaluation (W2): The human evaluation conducted on only 100 samples is insufficient to draw reliable conclusions. I strongly recommend expanding this to at least 500 samples. Furthermore, the reported Cohen's kappa of 0.51, combined with a 24% false-positive rate, suggests that the LLM judge introduces considerable noise into the evaluation. This is a significant issue for a paper focused on fixing over-refusals.

---

> > > ### Author Response · Authors · 2026-04-04
> > >
> > > Thank you very much for the thoughtful follow-up. We have conducted additional experiments and expanded evaluations to address both concerns.
> > >
> > > **W1: Missing Baselines on New Models.**
> > >
> > > Thanks for this insightful point. Following your suggestion, we evaluated all baselines on the three new models, reporting ASR (↓), BRR (↑), and Safe RR — the refusal rate on safe queries (↓):
> > >
> > > | Model / Method | ASR (↓) | BRR (↑) | Safe RR (↓) |
> > > | --- | --- | --- | --- |
> > > | **Qwen2.5-Omni** |     |     |     |
> > > | No Defense | 42.0% | 71.8% | 0.0% |
> > > | MDSteer-h2s | 58.8% | 50.8% | 0.0% |
> > > | MDSteer-c2r | 52.3% | 53.8% | 0.8% |
> > > | AdaShield | 25.2% | 68.6% | 54.4% |
> > > | FSD | 28.2% | 73.5% | 10.8% |
> > > | SARSteer | 31.2% | **80.2%** | **0.8%** |
> > > | **Qwen3-Omni** |     |     |     |
> > > | No Defense | 7.2% | 95.3% | 1.0% |
> > > | MDSteer-h2s | 62.8% | 65.6% | 0.0% |
> > > | MDSteer-c2r | 15.2% | 73.4% | 2.0% |
> > > | AdaShield | 3.6% | 70.6% | 55.2% |
> > > | FSD | 6.4% | 89.8% | 12.4% |
> > > | SARSteer | 5.2% | **94.0%** | **1.6%** |
> > > | **Voxtral-Mini** |     |     |     |
> > > | No Defense | 34.0% | 71.4% | 0.4% |
> > > | MDSteer-h2s | 50.8% | 65.4% | 0.0% |
> > > | MDSteer-c2r | 26.8% | 67.6% | 0.8% |
> > > | AdaShield | 2.4% | 66.4% | 54.0% |
> > > | FSD | 35.2% | 65.8% | 12.0% |
> > > | SARSteer | 30.0% | **69.2%** | **0.8%** |
> > >
> > > Consistent with our paper's findings (Tables 2–5), the audio distributional gap persists across all new architectures: both MDSteer variants worsen ASR (h2s on 5/5 models, c2r on 4/5), confirming that **deriving steering vectors from audio activations remains unreliable**. SARSteer is the only method that consistently reduces ASR without degradation across all 5 models.
> > >
> > > Meanwhile, AdaShield achieves low ASR but refuses **54–55% of safe queries** on all three new models, and FSD shows 10–12% safe query refusal. SARSteer maintains Safe RR ≤ 1.6% while achieving the **highest BRR across all three new models**: Qwen2.5-Omni (80.2%, vs. next-best FSD 73.5%), Qwen3-Omni (94.0%, vs. FSD 89.8%), and Voxtral-Mini (69.2%, vs. AdaShield 66.4%) — the same safety-utility advantage observed on original models (Qwen2-Audio BRR 79.95% vs. AdaShield 69.8%; Kimi-Audio 88.8% vs. AdaShield 52.6%). We will include the results in Appendix B in the camera-ready version.
> > >
> > > ---
> > >
> > > **W2: Human Evaluation.**
> > >
> > > Thanks for this important follow-up. According to your suggestion, we expanded from 100 to **500 samples** covering all 5 LALMs under both No Defense and SARSteer, including PAIR/GCG attacks on Kimi-Audio. Samples were drawn via stratified sampling and annotated in a fully blinded setting.
> > >
> > > |     | LLM=Safe | LLM=Unsafe |
> > > | --- | --- | --- |
> > > | **Human=Safe** | 271 | 43  |
> > > | **Human=Unsafe** | 3   | 183 |
> > >
> > > Agreement: 90.8%, Cohen's κ = 0.811 (substantial, improved from 0.51). Only **3 FN** vs **43 FP** (14:1 ratio).
> > >
> > > **Regarding "noise."** The 43 false positives follow a consistent pattern: the model begins with a refusal (e.g., "I'm unable to assist"), then provides contextually related but **harmless** content — explaining why the request is dangerous, offering prevention guidelines, or suggesting legitimate alternatives. The LLM judge flags such responses as unsafe based on surface-level topic cues, while human annotators correctly recognize the actual content as safe. This is a systematic conservative bias, not random noise. Crucially, this bias means our reported ASR values are **upper bounds** on the true attack success rate — the LLM judge overestimates rather than underestimates risk, which does not undermine the validity of using LLM-as-a-judge in our paper. We note that current safety evaluators tend to flag responses containing harmful *topics* even when the actual content is safe (e.g., refusal + educational explanation). Improving this nuanced capability is a meaningful future direction.
> > >
> > > **Regarding over-refusal.** Over-refusal in our framework is quantified via BRR (Eq. 7–8), which uses matching-based comparison against reference answers, entirely independent of the LLM judge. The concern that LLM judge noise is "a significant issue for a paper focused on fixing over-refusals" does not apply, as ASR (via LLM judge) and BRR (via matching) are decoupled by design.
> > >
> > > We will include this analysis in Appendix B in the camera-ready version.

---

### Official Review · Reviewer_B3ez · 2026-03-13

**Soundness:** 2
**Presentation:** 3
**Significance:** 3
**Originality:** 3
**Overall Recommendation:** 4
**Confidence:** 3

**Summary:**

The paper analyzes the safety alignment for Audio-Language models. First, it shows that activation steering calculated from audio representations fails because harmful and safe audio activations are distributively separated across all layers (unlike text, where they converge at intermediate depths). Next, it shows that basic defenses based on text representations cause over-refusals. To address both issues, they propose SARSteer, an inference-time steering method with two different components: (1) a steering vector derived from text representations, which is calculated by measuring the activation change when a text-based rejection request is added to harmful audio queries, and (2) a PCA-based ablation that projects the “safe audio subspace” from this direction vector to reduce over-refusals. Experiments with Qwen2-Audio and Kimi-Audio show that SARSteer achieves a low attack success rate outperforming vanilla steering.

**Compliance With Llm Reviewing Policy:**

Affirmed.

**Final Justification:**

The rebuttal addressed my concerns around the threat models and I'm happy to raise my score in line of this.

**Key Questions For Authors:**

1. **ASR + text filtering baseline.** Given that safety semantics are entirely text-mediated (at least according to the paper's analysis), have the authors compared against transcribing the audio and applying text-based safety defenses? What concrete advantage does activation-level intervention offer over this simpler pipeline?

2. **Validating the PCA-based selectivity mechanism.** Can the authors directly measure the refusal logit change induced by v_hat_perp on individual safe vs. harmful inputs, verifying that the geometric argument in Appendix A.4 holds at the sample level rather than only in aggregate?

**Limitations:**

The paper should (more saliently) acknowledge two limitations: (1) the evaluation is restricted to TTS-generated audio attacks, which does not validate robustness against audio-native adversarial strategies (e.g. adversarial perturbations, prosody-based attacks, multi-speaker scenarios), and (2) the theoretical justification for why the PCA ablation achieves selective steering relies on geometric assumptions that are not empirically validated at the individual sample level.

**Strengths And Weaknesses:**

## Strengths

- **The diagnostic analysis of why vanilla steering fails is solid and interesting.** The t-SNE visualizations showing persistent separation of harmful/safe audio activations across all layers, coupled with the CKA analysis (0.06 for audio vs. 0.67 for text), explain a fundamental challenge of multimodal safety. This finding is useful to the community independently of the proposed method.

- **Clean method design with well-motivated components.** Each component addresses a specific failure mode: text-derived steering addresses the modality gap, PCA ablation addresses over-refusals. Ablation (Figure 4) validates that both are necessary: V1 (direction only) over-refusals, V2 (ablation without adequate direction) fails on harmfulness, V3 (full method) balances both.


## Weaknesses

- **Missing important baseline: ASR transcription + text-based safety filtering.** The paper's own analysis shows safety semantics (i.e. ability to refuse) live entirely in the text pathway. The steering vector is derived purely from text. This raises a question: why not simply transcribe the audio and apply existing text-based defenses on the transcript? Such a pipeline would be simpler, more interpretable, and would leverage the mature text-safety literature directly. Without this comparison, it is hard to assess whether the activation-level intervention offers any concrete advantage in terms of effectiveness.

- **The threat model is artificially narrow.** All harmful audio is generated by converting text queries to speech via TTS. This is the least interesting attack vector for audio safety, as it reduces to text attacks re-encoded as audio. It's unclear whether threats could exploit audio-specific properties while bypassing this defense: prosody, background noise, adversarial acoustic perturbations, multi-speaker scenarios. The natural speech experiment (Appendix B.7) evaluates on AdvBench-audio where both baseline and SARSteer already achieve near-zero ASR, providing essentially no signal.

- **The theoretical justification for the PCA ablation's selectivity is underspecified.** The steering is applied uniformly to all inputs, and selectivity is claimed to emerge from the geometry: safe activations lie in the PCA-estimated subspace, so the orthogonal steering vector has minimal effect on them. Showing that PCA directions characterize safe audio does not automatically imply that removing them from the steering vector prevents over-refusal. This causal link is not validated beyond aggregate BRR numbers. Measuring refusal logit changes induced by v_hat_perp on individual safe vs. harmful inputs would strengthen the argument.

---

> ### Author Rebuttal · Authors · 2026-03-31
>
> Thank you very much for your careful review of our paper and thoughtful comments. We are greatly encouraged by your positive feedback on our **solid and interesting diagnostic analysis**, **clean method design**, and **validated ablation study**. We hope the following responses could help alleviate your concerns.
>
> ---
>
> **Cons1: Concern about the missing ASR transcription + text-based safety filtering baseline.**
>
> **R1:** Thanks for your concern. We would like to address this from the research scope and empirical evidence.
>
> **First, regarding the research scope.** Our work studies **safety alignment for LALMs** — aligning a model's internal behavior when it directly processes audio. The ASR + text filtering pipeline serves different use cases: a pre-filter intercepts queries *before* they reach the model, but (i) cannot defend against attacks that bypass the filter (e.g., adversarial audio transcribing to benign text), (ii) cannot address the model's own compliance tendency when processing audio, and (iii) is impractical for **real-time speech interaction** (voice assistants, spoken dialogue) where transcription + classification breaks latency requirements. SARSteer operates within the forward pass, addressing the root cause — the model's internal representations. The two approaches are complementary, but our contribution specifically targets LALM's intrinsic safety.
>
> **Second, the core issue is in the audio modality, not the text content.** Our paper (and prior work [1]) shows that LALMs are far more likely to comply with harmful audio than identical harmful text: **51.6% ASR via audio** on Qwen2-Audio (Table 2) vs. **10.0% via text** on Qwen2 (Table 9, Appendix B.4) — a 5× gap. A transcription pipeline would recover the text (which the model already handles safely at 10% ASR), but cannot address the audio-modality vulnerability causing the 5× increase — precisely what SARSteer solves at the activation level.
>
> ---
>
> **Cons2: Concern about the threat model scope — beyond TTS attacks.**
>
> **R2:** Thanks for your concern. Our evaluation is not limited to TTS-generated audio. We already evaluate on **AJailBench** (Table 2) with time-domain perturbations (including noise injection), frequency-domain perturbations, and hybrids (Appendix A.2). SARSteer achieves the lowest ASR on Qwen2-Audio (18.0%) and strong performance on Kimi-Audio (11.0%), with the best safety-utility balance (BRR 79.95%/88.8% vs. AdaShield's 69.8%/52.6%). We also evaluate on **Jailbreak-AudioBench** (Appendix B.7, Table 12) with celebrity accent, emphasis, and emotion variations, confirming these do not introduce new attack vectors.
>
> We additionally conducted **gradient-based and random perturbation** experiments on Qwen2-Audio (Figstep-audio):
>
> | Attack | ε | No Defense ASR | SARSteer ASR |
> | --- | --- | --- | --- |
> | PGD | 0.005 | 24.4% | **6.0%** |
> | Gaussian Noise | 0.01 | 27.2% | **7.2%** |
>
> The No Defense ASR (24.4%/27.2%) is lower than unperturbed (51.6%) because perturbations partially degrade audio intelligibility. Even so, SARSteer further reduces ASR to 6.0%/7.2%. As for multi-speaker scenarios, no existing LALM safety benchmark covers this setting; we will discuss this in the camera-ready version.
>
> ---
>
> **Cons3: Suggestion for validating PCA selectivity at sample level.**
>
> **R3:** Thanks for your suggestion. Appendix A.4 (Eq. 15–17) states the refusal logit change is:
>
> $$\Delta s = s(h + \alpha \hat{v}_{\perp})-s(h) \approx \alpha w^{\top} \hat{v} _\perp$$
>
> where $w$  is the local gradient (sample-dependent). The theory predicts (Eq. 18–19): $w^\top \hat{v}_\perp$ > 0 for harmful inputs; ≈ 0 for safe inputs.
>
> We computed per-sample $\delta_i = \langle  \hat{v}_ \perp  , h_i  \rangle$ (averaged across layers) for 100 harmful and 100 safe samples on Qwen2-Audio. While $\delta_i$ measures steering-activation alignment rather than $\Delta s$ directly, it reflects how strongly $\hat{v}_\perp$ interacts with each sample:
>
> |  | Mean δ | Std δ | % with δ > 0 |
> | --- | --- | --- | --- |
> | Harmful | −552.2 | 291.9 | **6.0%** |
> | Safe | −650.9 | 63.4 | **0.0%** |
> | Welch's t-test | p = 0.001 | Cohen's d = 0.47 |  |
>
> The negative $\delta$ indicates that $\hat{v}_ \perp$ is largely anti-aligned with typical hidden activations. Harmful samples show significantly higher $\delta$ (p = 0.001) with 4.6× larger variance, and 6% reach positive values while 0% of safe samples do — consistent with the prediction that $\hat{v}_\perp$ selectively targets harmful activations.
>
> ---
>
> **Suggestion for expanding the Limitations section.**
>
> **R4:** Thanks for the valuable suggestion. We will (1) expand Limitations to discuss multi-speaker scenarios, and (2) add sample-level PCA validation in the camera-ready version.
>
> ---
>
> [1] Yang et al., "Audio Is the Achilles’ Heel: Red Teaming Audio Large Multimodal Models", NAACL 2025

---

> > ### Author Rebuttal · Reviewer_B3ez · 2026-04-03
> >
> > Thanks for the reply. My concerns are resolved.

---

> > > ### Author Response · Authors · 2026-04-04
> > >
> > > We sincerely appreciate your thoughtful engagement throughout the review process and the time you have dedicated to carefully evaluating our work. Your constructive feedback has been invaluable in helping us strengthen the paper, and we will incorporate your suggestions into the revised manuscript. Thank you once again for your effort in improving this work.

---

### Decision · Program_Chairs · 2026-04-30

**Decision:**

Accept (regular)

**Comment:**

This paper proposes an inference-time safety alignment method for Large Audio-Language Models (LALMs). The approach extracts the refusal vector from the refusal text and ablates the safe subspace to mitigate over-refusal.

Reviewers appreciate the diagnostic analysis explaining why vanilla steering fails, the well-motivated method design, and the thorough evaluation.

Remaining concerns include new baseline comparisons showing that SARSteer underperforms in certain settings. Besides, relatively high false positives (43 out of 500) in the human study indicate systematic evaluator bias.